# Specific Targeting of PEGylated Liposomal Doxorubicin (Doxil^®^) to Tumour Cells Using a Novel TIMP3 Peptide

**DOI:** 10.3390/molecules26010100

**Published:** 2020-12-28

**Authors:** Mohammed S. Aldughaim, Munitta Muthana, Fatimah Alsaffar, Michael D. Barker

**Affiliations:** 1Research Center, King Fahad Medical City, P.O. Box 59046, Riyadh 11525, Saudi Arabia; 2Department of Oncology and Metabolism, University of Sheffield Medical School, Beech Hill Rd, Sheffield S10 2RX, UK; m.muthana@sheffield.ac.uk (M.M.); falsaffar@gc.edu.sa (F.A.); m.barker@sheffield.ac.uk (M.D.B.); 3Department of Clinical Laboratory Sciences, Alghad International Colleges for Applied Medical Sciences, Dammam 32423, Saudi Arabia

**Keywords:** angiogenesis, doxorubicin, TIMP3, VEGFR2

## Abstract

Doxorubicin is a cytotoxic anthracycline derivative that has been used as a chemotherapeutic in many different forms of human cancer with some success. However, doxorubicin treatment has several side-effects, the most serious of which is cardiomyopathy, that can be fatal. Doxorubicin encapsulation in PEGylated liposomes (Doxil^®^) has been shown to increase tumour localisation and decrease cardiotoxicity. Conversely, the stability of such liposomes also leads to increased circulation times and accumulation in the skin, resulting in palmar planter erythrodysesthesia, while also limiting release of the drug at the tumour site. Specific targeting of such liposomes to tumour cells has been attempted using various receptor-specific peptides and antibodies. However, targeting a single epitope limits the likely number of tumour targets and increases the risk of tumour resistance through mutation. In this report, Doxil^®^ was coupled to peptide sequence p700 derived from tissue inhibitor of metalloproteinase 3. This Doxil^®^ -P700 complex results in an approximately 100-fold increase in drug uptake, relative to Doxil^®^ alone, by both mouse and human breast cancer cells and immortalised vascular cells resulting in an increase in cytotoxicity. Using p700 to target liposomes in this way may enable specific delivery of doxorubicin or other drugs to a broad range of cancers.

## 1. Introduction

Doxorubicin is a well-established cytotoxic anthracycline derivative that has been used to successfully treat many different forms of human cancer, including breast cancer, soft tissue sarcomas, multiple myeloma and some leukaemia’s. Doxorubicin’s mechanism of action primarily involves intercalation of both chromosomal and mitochondrial DNA and inhibition of topoisomerase II, an enzyme that relaxes the DNA making it accessible to DNA polymerase, thus inhibiting DNA replication. However, doxorubicin binds to many cellular proteins in addition to topoisomerase, and also results in free radical production, increasing DNA damage and activation of apoptosis (reviewed in [1]). However, the application of the drug is limited due to a wide range of serious side-effects, such as liver and kidney damage and cardiotoxicity [2], and these restrict dosing regimens and, therefore, potential efficacy.

Novel drug delivery systems have since been employed in an attempt to limit such toxicity and improve targeting of these drugs. For instance, there is good evidence that encapsulation of cytotoxic drugs in liposomes helps alleviate the side effects and at the same time improves delivery [3,4,5]. In fact, the encapsulation of doxorubicin in PEGylated (polyethylene glycol-coated) liposomes (e.g., Doxil^®^/Caelyx^®^) has been demonstrated to improve its delivery and minimise side effects [6,7] by decreasing uptake by the mononuclear phagocyte system in the liver and spleen, greatly increasing circulation time and decreasing cardiotoxicity. Localisation is dependent on micelle size, preventing penetration of normal tissue, but allowing accumulation in cancer tissue with its abnormal, leaky vasculature. This is known as the enhanced permeability and retention (EPR) effect of tumours [8]. However, the stability of PEGylated liposomes can result in slow release of the liposomal content around the tumour, reducing cytotoxicity to the extent that there may be no net benefit over non-encapsulated drugs [9]. In addition, drugs released extracellularly into the tumour microenvironment may not be able to overwhelm the pumping action of multidrug resistant transporters. Moreover, one of the side-effects of Doxil^®^ is palmar planter erythrodysesthesia (PPE or hand-foot syndrome, a skin condition that particularly affects these limbs), probably due to increased circulation times resulting in accumulation of liposomes in capillaries below the skin [10]. This could possibly explain why liposomal doxorubicin only has a marginal effect on tumours that are refractory to most conventional chemotherapeutics—the exception to this being the AIDS-related skin cancer, Kaposi’s sarcoma, where accumulation in the skin is advantageous [11].

For this reason, modified liposomal formulations have been investigated that might enable more specific targeting of the drug to the tumour microenvironment and enhance uptake. These include the addition of peptides and antibodies that can serve as ligands to specific receptors or antigens that are uniquely expressed by tumour cells [12,13,14], as well as cytosolic/nuclear membrane penetrating peptides, such as those derived from the human immunodeficiency virus type-1 (HIV-1) transactivating transcriptional factor (TAT) protein, which contains an oligoarginine sequence that facilitates cellular entry [15].

We have previously identified a short 16 amino acid fragment of the C-terminus of TIMP3, designated as p700. This short peptide is present in loops 4 and 5 of full length TIMP3 that potently inhibits VEGFR2 [16]. In addition, p700 exhibits a broader binding specificity for VEGFR1, VEGFR3, PDGFRα, FGFR1, FDFR2α, FDFR3, and FDFR4. Interestingly, VEGFR2 and VEGFR3 are highly expressed within the tumour vasculature as well as in malignant breast, lung and colorectal tumours [17]. VEGFR1 is highly expressed in breast, prostatic, pancreatic and colorectal cancers [18,19,20], in addition to melanoma and leukaemia [21,22]. Additionally, FGF and PDGF receptors have been shown to be upregulated in tumours and/or tumour vasculature in colorectal, prostate, breast, ovarian, lung, and pancreatic cancers [23,24,25].

While it has been shown that it is possible to directly couple doxorubicin to cancer targeting peptides via a hydrolysable cross-linker, increasing specificity for cancer cells [26], there are several disadvantages of this approach. Firstly, modification in this way requires quite complex chemistry; secondly, modification of the drug may impair efficacy and, finally, the number of molecules of doxorubicin delivered will be equal to the number of target receptors, potentially reducing dosage to below the therapeutic threshold. This approach also assumes that the targeting peptide is internalised by the cells, and this was not known for p700.

An alternative strategy would be to couple p700 to the surface of a liposomal encapsulated doxorubicin, such as Doxil^®^. This has several potential advantages over direct coupling. Firstly, the chemistry is potentially more straight-forward and does not modify the doxorubicin itself. Secondly, one receptor can potentially enable delivery of many molecules of doxorubicin. Thirdly, each liposome can be coated in many copies of the peptide, potentially greatly enhancing avidity to the cell. Lastly, even if the p700 peptide is not internalised by the cell, localisation of the liposomes to the cell surface may still enable delivery of a therapeutic dose of Dox.

This study aimed to couple p700 to the surface of Doxil^®^ and determine the effect of coupling on binding, internalisation and cytotoxicity of the Doxil-p700 complex, relative to Doxil alone, by both human and mouse breast cancer cell lines and by primary human dermal microvascular endothelial cells and a mouse endothelial cell line.

## 2. Materials and Methods

### 2.1. H5V, 4T1 and MCF-7 Cell Culture

H5V cells are derived from mouse cardiac endothelial cells and express both VEGFR1 and VEGFR2. In particular, the phosphorylation of VEGFR2 in response to VEGF can readily be detected [27]. 4T1 cells are derived from a mouse mammary gland tumour and when grafted into BALB/c mice, represent a well-established animal model of metastatic human breast cancer [28,29]. MCF-7 cells are derived from human oestrogen and progesterone receptor positive invasive ductal carcinoma cells and have low metastatic potential.

H5V and 4T1 cell lines were grown in DMEM high glucose (4.5 g/L) media (for H5V) containing 1:1 potassium penicillin/streptomycin sulphate solution (stock 10,000 U/mL) at 10 µL/mL, 10% FBS, 2 mM L-glutamine and 56 mM of sodium bicarbonate (NaHCO_3_) at 37 °C and 5% CO_2_. MCF-7 cells on the other hand were cultured in Dulbecco’s Modified Eagle Medium (DMEM-F12) that was supplemented with 10% FBS, 1% penicillin/streptomycin, non-essential amino acids (0.1 mM), Insulin (10 µg/mL), and Sodium pyruvate (1 mM).

Primary human dermal microvascular endothelial cells (HuDMECs) used in this study were obtained from PromoCell, Heidelberg, Germany. HuDMEC cells were grown in endothelial cell growth medium, supplemented with 5% FCS, 1 µg/mL hydrocortisone, 10 ng/mL EGF and 0.4% Endothelial Cell Growth Supplement/Heparin (ECGS/H) at 37 °C and 5% CO_2_.

### 2.2. Conjugation of p700 Peptide to PEGylated Liposomal Doxorubicin

Dibenzocyclooctyl (DBCO)-functionalised, PEGylated-liposomal doxorubicin (Doxil^®^) was purchased from Encapsula^®^ Nanomedicines (Brentwood, TN, USA). Empty DBCO-functionalised liposomes (without doxorubicin) were also purchased from the same supplier for use as a negative control. Approximately 1% of the lipid in the liposomes comprised 1,2-distearoyl-sn-glycero-3-phosphoethanolamine (DSPE)-PEG-DBCO with the molarity of DBCO groups in the stock solutions approximately 220¦ÌM. In the liposomes containing the active drug, (Doxil^®^), the doxorubicin concentration was 2 mg/mL. Stock solutions were stored at 4 °C for up to one month.

Custom synthesis of an azide-functionalised, FITC-labelled p700 peptide, with a twelve-carbon polyethylene glycol spacer (p700-FITC), and an identical control but lacking the p700 peptide sequence (control-FITC), was carried out by Cambridge Research Biochemicals (Cambridge, UK) (see Table 1). Both p700-FITC and control-FITC were dissolved in water to make 3.3 mM stock solutions. Stocks were then stored at −20 °C.

p700-FITC or control-FITC were coupled to DBCO-Doxil^®^ or empty DBCO liposome by mixing 250 μL (55 × 10^−9^ moles DBCO) liposome stock and 50 μL (165 × 10^−9^ moles) of the azido-peptide stock and allowing the reaction to proceed with gentle agitation for at least 4 h at room temperature and then overnight at 4 °C in the dark.

Purification of the coupled products was carried out by cellulose ester membrane dialysis using the Float-a-Lyzer G2 (Spectrum^®^Labs, Rockleigh, NJ, USA,) with a 100 KDa molecular weight cut off, according to the manufacturer’s instructions. Dialysis was performed at 4 °C against three changes of 1 L PBS after 4, 6, and 12 h. Final volumes following dialysis were variable and made up to 500 µL with PBS. Based on a starting volume of Doxil of 250 μL, and assuming all the DBCO groups reacted with the azido-peptide, the final concentration of p700 in the final coupled stock was estimated to be 110 μM, with a final doxorubicin concentration of 1 mg/mL (1.7 mM). Purified stock conjugate solutions were stored at 4 °C in the dark and used within 2 weeks.

### 2.3. Confirmation of Coupling by Flow Cytometry

Flow cytometry was used to confirm coupling of FITC-labelled peptide to liposomal doxorubicin. Briefly, 30 µL of the post-dialysis sample was diluted ten-fold with PBS and assayed using a BD LSRII flow cytometer. Samples were excited using the blue laser at an excitation wavelength of 488 nM and emissions of 530 nM and 575 nM were used to detect green fluorescence (FITC) and red auto-fluorescence (doxorubicin), respectively. Uncoupled, empty liposomes were used to set the negative values for both channels, and uncoupled Doxil^®^ was used as a positive control for the red channel and p700-FITC as a positive control for the green channel. Data were analysed using FlowJo software.

### 2.4. Confirmation of Cellular Binding of p700-Conjugated Liposomal Doxorubicin

Cellular binding of the p700-conjugated liposomal doxorubicin to the human and mouse breast cancer cell lines MCF-7 and 4T1 (ATCC, Teddington, UK), respectively, was evaluated by flow cytometry (refer to Appendix A). The mouse cardiac endothelial cell line, H5V (a kind gift from Dr Annunciata Vecchi, Istituto Clinico Humanitas, Rozzano, Italy), which expresses high levels of VEGFR2 [27] was used as a positive control, and primary human dermal microvascular endothelial cells (HuDMEC–PromoCell, Heidelberg, Germany), were used to determine uptake by normal endothelial cells. Cells were cultured overnight in a 6-well plate at a density of 5 × 10^4^ per well for H5V, MCF-7 and 4T1, and 1 × 10^5^ per well, for HuDMECs (due to their longer doubling time). To each well containing 2 mL culture media, 5 μL of either Doxil^®^-p700-FITC or Doxil-FITC stock solution (i.e., 275 nM peptide) was added and incubated at 37 °C for 4 h in the dark. The media was then removed and the cells washed once with PBS.

To detach cells, 800 µL of Accutase (SIGMA, UK, Gillingham, Dorset) was then added to each well and incubated for 2–3 min at 37 °C. Culture media (800 µL) was added to stop the Accutase action and the cell suspension transferred to a polystyrene falcon tube and spun for 5 min at 500× *g*. The supernatant was removed and cell pellet resuspended in 300 µL FACS buffer (Dulbecco’s phosphate buffer saline (DPBS) containing 0.2% (*w*/*v*) bovine serum albumin and 0.1% (*w*/*v*) sodium azide).

The samples were then examined on an LSR II flow cytometer using compensated-blue laser at excitation wavelength of 488 nM and emissions of 530 nM with 10,000 events collected. Using FlowJo software, data from treated and untreated cells were plotted as side incidental light scatter (SSC) against green fluorescence. For each sample, the percentage of cells with green fluorescence intensity above that of untreated cells was then determined.

### 2.5. Confirmation to Cellular Uptake of p700-Conjugated Liposomal Doxorubicin

Cellular uptake and localisation of the p700-conjugated liposomal doxorubicin (Doxil^®^-p700-FITC) by MCF-7, 4T1, H5V and HuDMEC cells was evaluated by confocal microscopy. Cells were cultured overnight on a 4-well glass chamber slide (Nunc Lab-Tek^®^II, Thermo Fisher Scientific, UK, Loughborough) at a density of 3 × 10^4^ per well (for H5V, MCF-7 and 4T1) and 5 × 10^4^ per well (for HuDMECs). To each well containing 700 µL media, 1.75 µL of Doxil^®^-p700-FITC; Doxil-FITC or empty liposome-p700-FITC (lacking doxorubicin) stock solution was added (equivalent to 275 nM peptide in each case). The cells were then incubated at 37 °C in the dark for 4, 9 or 24 h. The media was then removed and the cells washed once with HBSS to remove any phenol-red containing media. Cells were subsequently fixed by incubating for 20 min at room temperature with 500 µL per well of 4% paraformaldehyde (diluted in HBSS).

Fixed cells were then washed twice in PBS followed by a 5-min incubation in 0.1 M glycine (pH 7.4) at room temperature. They were then washed three times in PBS and their nuclei stained with 4′,6-diamidino-2-phenylindole (DAPI 50 μg/mL in PBS) by incubation for 1–2 min at room temperature. Another wash with PBS was then performed and the plastic chamber then detached from the glass slide. Any excess PBS was removed and the slide mounted with Prolong Gold Antifade (Life Technologies). Once set, slides were examined on a laser scanning (Nikon A1) confocal microscope equipped with FITC (EX/EM 488/530 nM), Red (EX/EM 562/570 nM) and UV-Blue (EX/EM 405/480 nM) filters. Pictures were then further analysed using Image J software.

### 2.6. Cytotoxicity Assay

The cytotoxic effect of increasing concentrations of Doxil^®^-p700-FITC compared with Doxil-FITC and empty liposome-p700-FITC was evaluated using an MTS assay (CellTiter 96^®^ AQ_ueous_, Promega, UK, Southampton). Cultured 4T1, H5V and HuDMEC cells at around 80% confluence were trypsinised and then seeded at densities of 2 × 10^3^ per well containing 100 µL culture media in a 96-well plate and incubated at 37 °C overnight.

Cells were then treated with Doxil^®^-p700-FITC, Doxil-FITC, and empty liposome-p700-FITC at 0, 2.5, 5, 10, 25, 50, and 100 µM doxorubicin concentration (or equivalent liposome concentration for the empty liposomes) for 4 h at 37 °C, before being washed three times with fresh media to remove any unbound drug. They were subsequently incubated for a further 24 h in fresh media at 37 °C before an MTS assay was performed. Each dose level was performed in triplicate and three independent repeat experiments were completed.

Statistical analysis was performed using two-way ANOVA-multiple comparison test and percentage cell viability curves were drawn using GraphPad Prism software v7.0.

### 2.7. Statistical Analysis

Experiments were performed in triplicate and data is presented as mean ± SEM of three independent repeats (** = *p* < 0.01, *** = *p* < 0.001, **** = *p* < 0.0001 indicates significance, one or two-way ANOVA, multiple comparison test).

## 3. Results

### 3.1. Coupling Assessment

Flow cytometry confirmed the coupling of the azido-FITC peptides to either empty DBCO-liposomes or DBCO-Doxil (Figure 1). As can be seen in panel (A) for azido-p700-FITC and panel (B) for azido-FITC only, there was a pronounced shift in the green fluorescence associated with the liposomes before (blue peak) and after (green peak) coupling, indicating efficient cross-linking of both azido-molecules to the liposomes. This result was mirrored for coupling of azido-p700-FITC and azido-FITC only (panels (C) and (D) respectively) to DBCO-Doxil. Although the background green fluorescence was slightly higher for the Doxil than the empty liposomes prior to coupling (red peak), the shift following coupling (green peak) was also more pronounced, again confirming efficient coupling of both azido-FITC molecules to DBCO-Doxil. Panels (E) and (F) compare the red fluorescence (associated with doxorubicin) of the final cross-linked products (blue peak, empty liposomes and red peak, Doxil), which confirmed doxorubicin was still present in the Doxil following coupling. Panels (E), for p700-FITC and (F), for FITC-only, compare the red (doxorubicin) fluorescence of the empty liposomes (blue peaks) versus Doxil (red peaks) following coupling, confirming the presence of doxorubicin in the coupled Doxil.

### 3.2. Binding Evaluation

Conjugates were then evaluated for binding to a panel of cell lines to determine whether p700 enhances binding of Doxil to mouse and human breast tumour cell lines (4T1 and MCF-7, respectively) or a mouse endothelial cell line (H5V) known to highly express VEGFR2, relative to human primary microvascular endothelial cells (HuDMEC). Doxil-FITC only was used as a negative control.

As shown in Figure 2A–C, 96–99% of the MCF-7, 4T1 and H5V cells treated with Doxil-p700-FITC showed positive FITC staining, compared to 3% or less of those cells treated with Doxil-FITC only. In contrast, only about 15% of HuDMEC showed positive FITC staining with Doxil-p700-FITC (Figure 2D), with those treated with the Doxil-FITC alone showing less than 2% positive FITC staining. This difference in staining between the cell lines and the primary microvascular endothelial cells was highly significant, as illustrated in Figure 2E.

### 3.3. Cellular Uptake and Internalisation

The purpose of coupling p700 to Doxil was to specifically target the cytoxic drug doxorubicin to tumour tissue. While flow cytometry confirmed that p700 greatly enhances binding of Doxil to tumour cells and endothelial cells that show up-regulation of target receptors (H5V), it is vital that the doxorubicin is then taken up by the cells in order to be effective. Localisation of Doxil to the cell surface may be sufficient to enhance doxorubicin uptake by the cells, however whether p700 would be internalised following binding and thus potentially facilitate cellular entry of the doxorubicin, was unknown. To examine cellular internalisation after binding, 4T1 cells were incubated with Doxil-FITC only or Doxil-p700-FITC for 4, 9 and 24 h, after which cellular internalisation of both peptide (as determined by the FITC label) and doxorubicin (as determined by doxorubicin fluorescence) were evaluated using confocal microscopy (Figure 3). In the absence of the p700 sequence, no FITC fluorescence was seen to be associated with the cells (Figure 3A) although there was some red fluorescence, presumably due to passive uptake of doxorubicin. In contrast, in cells treated with Doxil-p700 (Figure 3B), there was very strong FITC fluorescence, largely associated with what are probably endosomal granules in the cytoplasm, and this co-localised with strong red fluorescence, presumed to be from the doxorubicin (observed as a yellow colour in the merged images). Florescence was maximal at the 9-h time point, becoming somewhat diminished at 24 h. The fact that the red fluorescence was due to doxorubicin was confirmed by repeating the experiment with the empty liposomes coupled to p700 (Figure 3C), where only green fluorescence was observed.

The experiment was then repeated for H5V, MCF-7 and HuDMEC (Figure 4), however in this case only data for 9 h are shown as maximal fluorescence was seen at this time point. Again, for H5V and MCF-7 (Figure 4A,B respectively), the p700 sequence greatly enhanced uptake of FITC and doxorubicin, with no green and little red fluorescence observed in the cells treated with Doxil-FITC only. However, for HuDMEC, no detectable green or red fluorescence was observed with either Doxil-FITC or Doxil-p700-FITC, suggesting little internalisation of the drug (Figure 4C).

### 3.4. Cytotoxic Effect of Conjugates

Cytotoxicity of the conjugates towards MCF-7, 4T1, H5V and HuDMEC was determined 24 h after a 4-h exposure to varying concentrations of Doxil-p700-FITC or Doxil-FITC. Empty liposome-p700-FITC was used as a control to confirm any cytotoxicity seen was due to the doxorubicin content of the Doxil and not the liposome or p700 peptide. As shown in Figure 5, 4T1, MCF-7 and H5V cells treated with the highest dose of Doxil-p700-FITC all showed a significant increase in cell death, relative to both empty p700-liposomes and Doxil-FITC only. While the HuDMECs also showed a significant increase in cell death with Doxil-p700, relative to the empty p700-liposomes, this was not significantly greater than that seen with Doxil-FITC only.

## 4. Discussion

The data reported herein confirm that p700 retains its ability to target tumours and tumour vasculature via tyrosine kinase receptors when covalently cross-linked to PEGylated liposomes and that this clearly enhances uptake of Doxil into these cells. However enhanced internalisation does not necessarily equate to enhanced cytotoxicity if the doxorubicin is retained in endosomes and cannot reach the nucleus. After internalisation of liposomal drugs, the liposome is usually metabolised in the endolysosomal pathway where the liposome is degraded by the acidic medium of the lysosome [30]. The reduction in Doxil-p700-FITC concentration in the cytoplasm after 9 h could be because of degradation of the peptide and the liposome. However, the fact that there was a clear enhancement of cytotoxicity of Doxil-p700 over Doxil only, for all the cancer cell lines at 100 µM, indicates that at least some of the doxorubicin is then released to the nucleus. Although it is clearly difficult to compare treatment of cells in vitro in a static culture with the in vivo situation, the maximum recommended clinical dose for Doxil is 50 mg/m^2^ [31], which would be in the order of 30 µM blood concentration of doxorubicin. As the p700-Doxil is expected to be highly concentrated within the tumour microenvironment, it does not seem unreasonable to expect a concentration of 100 µM to be achievable in this millieu.

Binding of Doxil-p700-FITC was clearly evident by the much higher percentage of FITC positive population in the second quadrant of, respectively, MCF-7 96.4%, 4T1 99.7%, and H5V 99.0% compared to primary HuDMEC, which showed only 15% positive cells for binding. This may be an indication of the highly expressed receptors on cancer cells that can bind p700, those which are less expressed in normal cells such as the case with HuDMEC.

Binding and uptake of Doxil-p700 by the human MCF-7 breast tumour cells was somewhat less than for the mouse 4T1 breast tumour cells and this may explain the reduced killing of MCF-7 relative to 4T1 tumours. However, MCF-7 is also known to be resistant to doxorubicin with specific point mutations shown to confer this cell line with resistance to anthracyclines [32]. Indeed, this is probably reflected in Figure 5, in which MCF-7 is the only cell line where Doxil alone is not cytotoxic, relative to the empty liposomes. The fact that Doxil-p700 does still show some killing of this cell line demonstrates the benefit of targeted delivery over conventional methods.

Perhaps surprisingly, uptake of Doxil-p700 by tumour cells was much greater than for primary human microvascular endothelial cells (HuDMEC), despite the fact that these cells do bear VEGFR2. In contrast uptake by the H5V mouse endothelial cell line was much higher. This may, in part, be due to the differential expression of VEGFR2 by these cells and could be said to mimic the fact that VEGFR2 is upregulated in tumour microvasculature compared to normal tissue. It is also possible that, as a retrovirally transformed cell line [33], H5V also bears FGF and/or PDGF receptors, which are not expressed significantly by HuDMEC. Despite this, Doxil-p700 did show significant killing of HuDMECs, relative to the empty liposomes. However, this probably reflects the fact that primary cells lack some of the drug resistance mechanisms of tumour cells and in fact the level of killing seen with Doxil-p700 was no greater than for Doxil alone, suggesting that the doxorubicin was largely internalised passively into these cells. Moreover, normal vasculature is unlikely to be exposed to such high concentrations of Doxil-p700 as the tumour tissue where it should be concentrated. Binding of the p700-Doxil conjugates to the cell surface was supposed to facilitate internalisation by the cells. It might be noticeable that the uptake result in the confocal image does not reflect the binding of the conjugate to cell surface in the flow cytometry. This is because only 15% of the cells showed binding to cell surface which would translate to reduced probability of internalisation or uptake of the conjugate in HuDMEC as seen by nearly no florescence. However, over 90% binding in the cancer cells resulted in significantly higher uptake of the conjugate.

In this study, we sought to exploit the multi-receptor binding profile of p700 [16] to target PEGylated doxorubicin (Doxil) to tumour cells. Although this is not a novel concept, p700 may offer several advantages over conventional peptides or antibodies. As far as we are aware, p700 is unique in being able to target multiple pro-angiogenic growth factor receptors known to be upregulated in tumours or tumour vasculature, giving it the potential to target a wide range of tumours with high avidity due to multivalent binding. Targeting multiple receptors also greatly decreases the likelihood of tumour cells developing resistance due to downregulation or mutation of any one receptor type. An alternative approach, previously used, has been to exploit cell-penetrating peptides such as those derived from TAT [15], although such peptides are not tumour specific and so risk increased off-target effects.

Usually, ligands targeting multiple receptors are often characterised with varying specificity for their targets [34], which is a likely problem for p700. However, the receptors inhibited by p700 are a small, closely related group of proangiogenic tyrosine kinase receptors that are not highly expressed by normal cells. Thus, p700 is expected to be highly specific to tumours. There are plenty of mono-specific molecules available but these are highly vulnerable to tumour drug resistance due to mutation. Likewise, there are much broader small molecule inhibitors that target several tyrosine kinases such as Sorafenib and Sunitinib, which have been approved for cancer treatment [35]. These molecules also inhibit some intracellular tyrosine kinases, potentially increasing off-target effects. The potential advantage of p700 is its specificity for a narrow range of extracellular tyrosine kinase receptors and this can be employed to deliver any payload that can be encapsulated in liposomes.

In addition to binding multiple targets, p700 also has several advantages over using monoclonal antibodies to target liposomes. The latter are expensive to develop and produce, require humanization to prevent an immune response and show decreased tissue penetration due to their large size. In contrast, p700 is only 16 amino acids long and is derived from a highly conserved secreted human protein, so that immunogenicity is much less likely to be an issue.

While there are several small molecule drugs that also target a wider family of tyrosine kinase receptors, such as sorafenib and sunitinib, these all target the intracellular kinase domain and can also target other intracellular kinases with the potential for off-target effects [36]. Using extracellular, competitive inhibitors of these receptors to target liposomes should decrease the likelihood of side-effects and enable the deliver any drug that can be encapsulated in this way.

## 5. Conclusions

The data presented here demonstrate that Doxil-p700 has significant potential as a cancer therapeutic with the ability to actively target doxorubicin to tumour tissue. Enhanced localisation to the tumour may decrease circulation time throughout the body, relative to unmodified Doxil, potentially reducing off target effects such as palmar planter erythrodysesthesia. Additionally, p700 enabled active internalisation of the conjugate, potentially enhancing drug delivery into the tumour cells. While some cells, such as MCF-7, show resistance to doxorubicin, the other advantage of this technique is that it can be used to target any drug that can be encapsulated in this way.

## Figures and Tables

**Figure 1 molecules-26-00100-f001:**
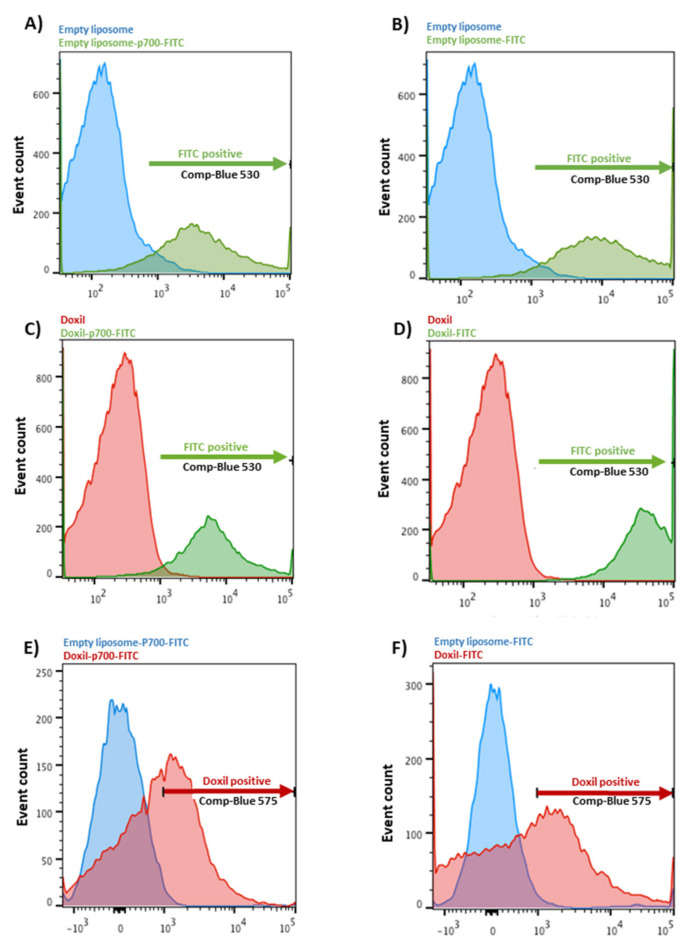
Assessment of p700 coupling to PEGylated liposomes. Flow cytometric analysis of coupling of: (**A**) empty DBCO-liposome to azido-p700-FITC; (**B**) empty DBCO-liposome to azido-FITC only; (**C**) DBCO-Doxil to azido-p700-FITC; (**D**) DBCO-Doxil to azido-FITC only. Panels (**A**–**D**) show the shift in green (FITC) fluorescence before (blue or red peaks) and after (green peaks) coupling. Panels (**E**), for p700-FITC and (**F**), for FITC-only, compare the red (doxorubicin) fluorescence of the empty liposomes (blue peaks) versus Doxil (red peaks) following coupling, confirming the presence of doxorubicin in the coupled Doxil.

**Figure 2 molecules-26-00100-f002:**
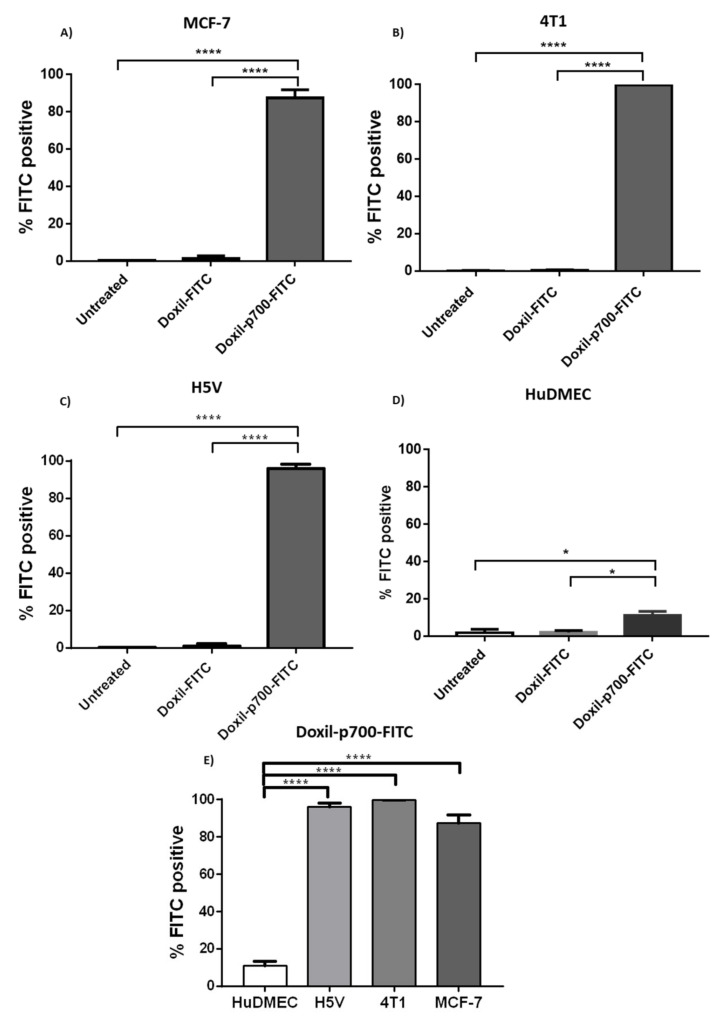
The effect of coupling p700 peptide on the binding of Doxil to cells. Bar charts showing comparative flow cytometric data for the binding of Doxil-FITC or Doxil-p700-FITC to (**A**) MCF-7; (**B**) 4T1; (**C**) H5V or (**D**) HuDMEC. (**E**) Relative binding of Doxil-p700-FITC to the above cell lines. Percentages relate to numbers of cells with a fluorescence intensity above that of untreated cells. Data are means ± SEM, n = 3 (**** *p* < 0.0001, * *p* < 0.01 indicates significance, one-way ANOVA, multiple comparison test).

**Figure 3 molecules-26-00100-f003:**
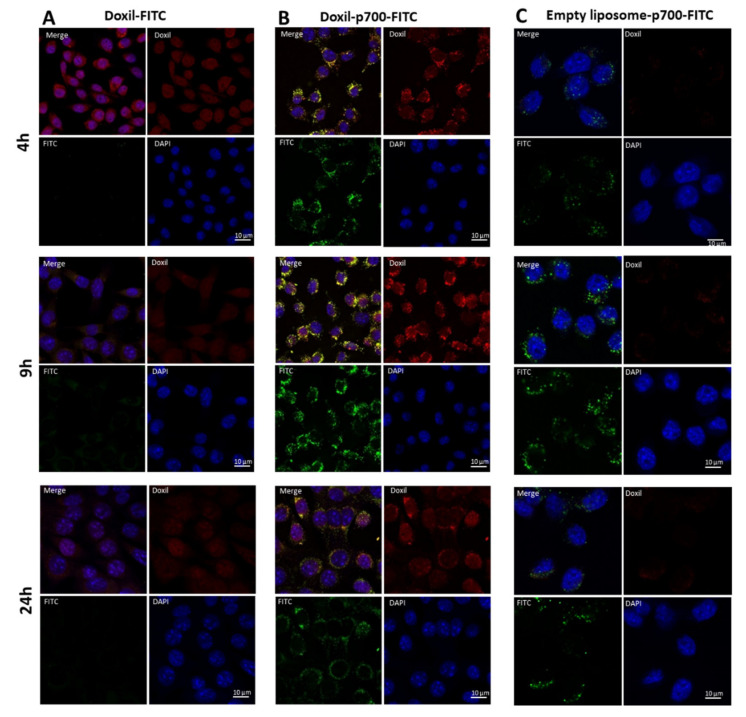
Uptake of Doxil, Doxil-p700 and empty liposome-p700 by the 4T1 cell line with time. 4T1 mouse breast cancer cells were treated with (**A**) Doxil-FITC; (**B**) Doxil-p700-FITC or (**C**) empty liposome-p700 at 275 nM for 4, 9 and 24 h at 37 °C and then imaged by confocal microscopy. Red staining is doxorubicin, green staining is FITC on the peptide conjugate and blue DAPI nuclear stain.

**Figure 4 molecules-26-00100-f004:**
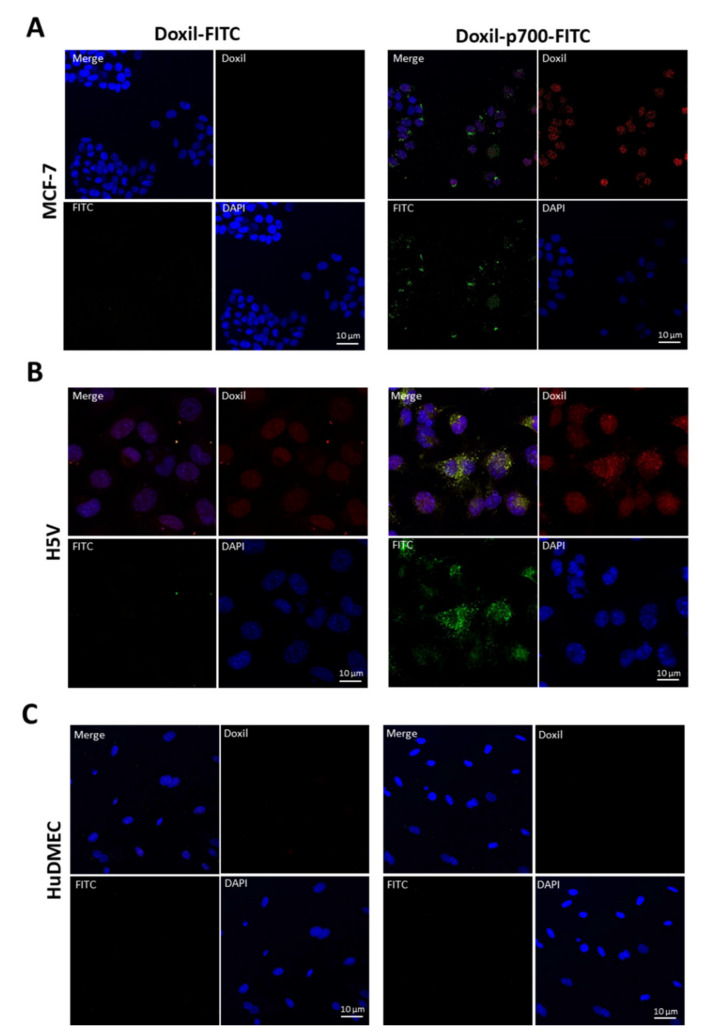
Uptake of Doxil and Doxil-p700 by H5V, MCF-7 and HuDMEC after 9 h. HV5 (**A**), MCF-7 (**B**) or HuDMEC (**C**) were treated with Doxil-FITC or Doxil-p700-FITC, as indicated, at 275 nM for 9 h at 37 °C and then imaged by confocal microscopy. Red staining is doxorubicin, green staining is FITC on the peptide conjugate and blue DAPI nuclear stain.

**Figure 5 molecules-26-00100-f005:**
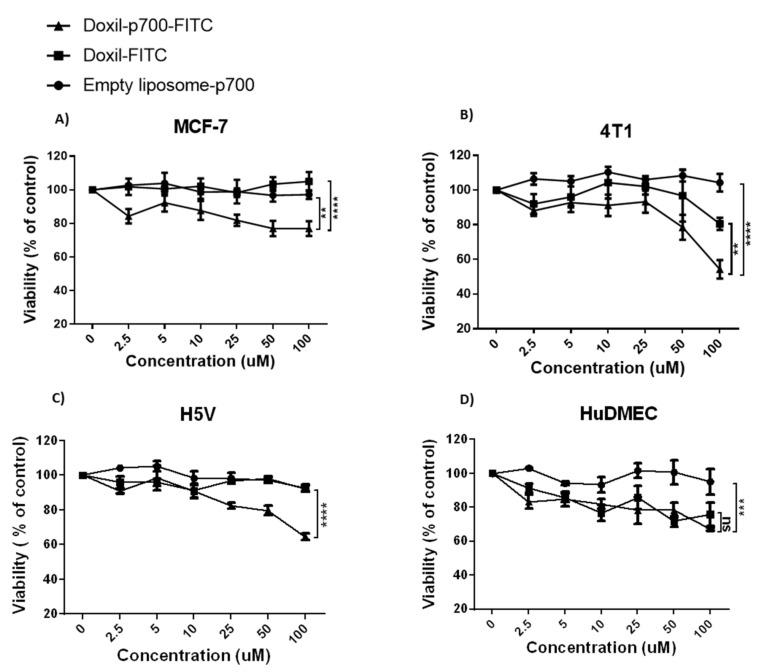
Effect of increasing concentrations of empty liposome—p700 FITC, Doxil-FITC and Doxil-p700-FITC on cell viability. 4T1 (**A**), MCF-7 (**B**), H5V (**C**) or HuDMEC (**D**) were treated for 4 h with the stated dose of each agent, as indicated in the key, washed and then incubated for a further 24 h in complete growth medium before performing an MTS cell viability assay. Data are means ± SEM, n = 3 (** = *p* < 0.01, *** = *p* < 0.001, **** = *p* < 0.0001 indicates significance, two-way ANOVA, multiple comparison test).

**Table 1 molecules-26-00100-t001:** Structure of p700-FITC and control FITC.

Abbreviation	Molecular Weight	Structure
Control-FITC	1159.5	Azido-PEG12-Lys(5-FITC)-amide
P700-FITC	3032.4	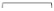 Azido-PEG12-KIKSCYYLPCFVTSKN-Lys(5-FITC)-amide

## Data Availability

The data presented in this study are available in Appendix A.

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
