# Peer review of "Specific Targeting of PEGylated Liposomal Doxorubicin (Doxil®) to Tumour Cells Using a Novel TIMP3 Peptide"

_molecules, 2020, doi:10.3390/molecules26010100_

Round 1
Reviewer 1 Report
Recommendation: Publish after minor revisions noted.
Comments:
This manuscript by Aldughaim reports the coupling of Doxil to peptide p700 to direct doxorubicin to cancer cells to avoid side-effects. The Doxil-p700 exhibits an approximately 100-fold increase in drug uptake by cancer cells. Doxil-p700-FITC selectively binds to cancerous cells over healthy cells. The work appears to have not been reported before. However, the following items should be addressed prior to publication:
- Abstract is too long and includes too much background information. Please summarize the background information in one or two sentences and shorten the abstract to one short paragraph, emphasizing what was done and what conclusion was obtained.
The author may consider presenting in this kind of direct way: "To avoid side-effects, like cariotoxicity, Doxil was coupled to peptide sequence p700 derived from tissue inhibitor of metalloproteinase 3. This Doxil-P700 complex results in an approximately 100-fold increase in drug uptake…"
- Line 100: H5V, 4T1, and MCF-7 [comma is needed after H5V]
- For audiences who are not familiar with click reaction, it would be helpful to include a figure/scheme showing the reaction between DBCO-Doxil and azido-p700-FITC. Also, the structure of Doxil-triazole-p700-FITC will help to visualize what is under investigation.
- Line 137: Separation of the coupled and uncoupled peptide and p Purification of the coupled products was carried out by…
- FITC at the Doxil®-p700-FITC, Doxil-FITC, and empty liposome-p700-FITC was employed for imaging purpose. However, for cytotoxicity of Doxil®-p700, FITC is not needed. Moreover, fluorescein is toxic and can cause adverse reactions. Thus, it would make this manuscript stronger if the author is able to also include the cytotoxicity of Doxil®-p700. Or the author can test Doxil®-p700 in the further for another manuscript.
- Explain what Fig 1. (E) and (F) are in the legend.
- Fig3. C should be Empty liposome-p700-FITC
8. A conclusion is needed
Author Response
Dear Sir or Madam:
(Specific targeting of PEGylated liposomal doxorubicin (Doxil®) to tumour cells using a novel TIMP3 peptide)
Thank you for the opportunity to respond to the referees’ comments regarding the above submitted manuscript. We will, however, attempt to address the referees’ comments as best we can in the response to reviewers’ comments below. These are shown in bold.
Reviewer 1
- Abstract is too long and includes too much background information. Please summarize the background information in one or two sentences and shorten the abstract to one short paragraph, emphasizing what was done and what conclusion was obtained.
Abstract has been updated as per suggested by the reviewer.
- Line 100: H5V, 4T1, and MCF-7 [comma is needed after H5V]
Comma has been added.
- For audiences who are not familiar with click reaction, it would be helpful to include a figure/scheme showing the reaction between DBCO-Doxil and azido-p700-FITC. Also, the structure of Doxil-triazole-p700-FITC will help to visualize what is under investigation.
Click chemistry reaction was clearly illustrated in the graphical abstract submitted. Please see attached file.
- Line 137: Separation of the coupled and uncoupled peptide and p Purification of the coupled products was carried out by…
Updated as per suggested.
- FITC at the Doxil®-p700-FITC, Doxil-FITC, and empty liposome-p700-FITC was employed for imaging purpose. However, for cytotoxicity of Doxil®-p700, FITC is not needed. Moreover, fluorescein is toxic and can cause adverse reactions. Thus, it would make this manuscript stronger if the author is able to also include the cytotoxicity of Doxil®-p700. Or the author can test Doxil®-p700 in the further for another manuscript.
Yes, that would be the plan for the upcoming work/manuscript.
- Explain what Fig 1. (E) and (F) are in the legend.
That’s true, thank you for highlighting this. Legend has been updated now by adding: Panels (E), for p700-FITC and (F), for FITC-only, compare the red (doxorubicin) fluorescence of the empty liposomes (blue peaks) versus Doxil (red peaks) following coupling, confirming the presence of doxorubicin in the coupled Doxil.
- Fig3. C should be Empty liposome-p700-FITC
Thank you. Figure has been updated now.
- A conclusion is needed
Yes, sure. A conclusion section has been added now.

Reviewer 2 Report
In this paper, the authors describe the synthesis of doxil loaded liposomes conjugated with the P700 peptide that binds TIMP3. The authors claim that the P700-conjugated liposomes show improved uptake in breast cancer cells and shows increased cytotoxicity. The results very convincingly show that Doxil-P700-FITC liposomes are taken up more effectively in breast cancer cell lines and shows better cytotoxicity compared to Doxil-FITC. Moreover, the Doxil-P700-FITC liposomes are not taken up effectively by primary human dermal microvascular endothelial cells. Overall, the results support the conclusions. A few minor points to consider:
- It might be useful to explain why the authors chose dermal microvascular endothelial cells to test for cytotoxicity on non-cancerous cells instead of cardiac microvascular endothelial cells since they are available from the same vendor. I understand that the H5V cells were meant to provide insights into the potential cardiotoxicity of the formulation. The do authors discuss the difference between H5V and HuDMECs in the discussion. However, since the Doxil-p700 formulation is active in H5V cells, I am wondering if the authors have any data on potential differences between HuDMECs vs cardiac microvascular endothelial cells and if they expect primary cardiac microvascular endothelial cells to behave similarly to HuDMECs.
- Secondly, the file titled “non-published” material provides data on the magnitude of uptake of Doxil-P700-FITC in the cells. It might be interesting to add this data to the figures at least in an abbreviated manner (either as text or e a bar chart listing the median fluorescence intensity)
Author Response
Dear Sir or Madam:
(Specific targeting of PEGylated liposomal doxorubicin (Doxil®) to tumour cells using a novel TIMP3 peptide)
Thank you for the opportunity to respond to the referees’ comments regarding the above submitted manuscript. We will, however, attempt to address the referees’ comments as best we can in the response to reviewers’ comments below. These are shown in bold.
Reviewer 2:
- It might be useful to explain why the authors chose dermal microvascular endothelial cells to test for cytotoxicity on non-cancerous cells instead of cardiac microvascular endothelial cells since they are available from the same vendor. I understand that the H5V cells were meant to provide insights into the potential cardiotoxicity of the formulation. The do authors discuss the difference between H5V and HuDMECs in the discussion. However, since the Doxil-p700 formulation is active in H5V cells, I am wondering if the authors have any data on potential differences between HuDMECs vs cardiac microvascular endothelial cells and if they expect primary cardiac microvascular endothelial cells to behave similarly to HuDMECs.
Thank you. As mentioned in the manuscript, H5V cells are derived from mouse cardiac endothelial cells and express both VEGFR1 and VEGFR2. In particular, the phosphorylation of VEGFR2 in response to VEGF can readily be detected. Uptake of Doxil-p700 by tumour cells was much greater than for HuDMEC, despite the fact that these cells do bear VEGFR2. In contrast uptake by the H5V mouse endothelial cell line was much higher. This may, in part, be due to the differential expression of VEGFR2 by these cells and could be said to mimic the fact that VEGFR2 is upregulated in tumour microvasculature compared to normal tissue. It is also possible that, as a retrovirally transformed cell line, H5V also bears FGF and/or PDGF receptors, which are not expressed significantly by HuDMEC. Despite this, Doxil-p700 did show significant killing of HuDMECs, relative to the empty liposomes. However, this probably reflects the fact that primary cells lack some of the drug resistance mechanisms of tumour cells and in fact the level of killing seen with Doxil-p700 was no greater than for Doxil alone, suggesting that the doxorubicin was largely internalised passively into these cells. Moreover, normal vasculature is unlikely to be exposed to such high concentrations of Doxil-p700 as the tumour tissue where it should be concentrated.
- Secondly, the file titled “non-published” material provides data on the magnitude of uptake of Doxil-P700-FITC in the cells. It might be interesting to add this data to the figures at least in an abbreviated manner (either as text or e a bar chart listing the median fluorescence intensity).
Thank you for the comment. Binding data (uptake) is already added in the original manuscript as supplementary data so it will be published as well.
Again, thank you for the opportunity to respond to all these valid comments.